Filling the gaps in ecology of tropical tiger beetles (Coleoptera: Cicindelidae): first quantitative data of sexual dimorphism in semi-arboreal Therates from the Philippine biodiversity hotspot

Acal Dale Ann dale.ann.acal@edu.uni.lodz.pl
Sulikowska-Drozd Anna
http://orcid.org/0000-0001-8949-848X Jaskuła Radomir
Department of Invertebrate Zoology and Hydrobiology, Faculty of Biology and Environmental Protection, University of Lodz , Łódź , Poland
Silva Daniel
Electronic publication date: 2024 Mar 13
Publication date: 2024
Volume: 12
Electronic Location ID: e16956
Received 2023 Oct 25; Accepted 2024 Jan 25
Copyright: © 2024 Acal et al.
Copyright year: 2024
Copyright holder: Acal et al.
License: This is an open access article distributed under the terms of the Creative Commons Attribution License, which permits unrestricted use, distribution, reproduction and adaptation in any medium and for any purpose provided that it is properly attributed. For attribution, the original author(s), title, publication source (PeerJ) and either DOI or URL of the article must be cited.
License URL: https://creativecommons.org/licenses/by/4.0/

Keywords: Adephaga, Insects, Sexual dimorphism, Body size variation, Philippines

Funding: Department of Science and Technology-Science Education Institute (DOST-SEI) DOST- Foreign Graduate Program University of Lodz statutory funds This research is supported by the Department of Science and Technology-Science Education Institute (DOST-SEI), Philippines. Dale Ann Acal received financial support from the DOST- Foreign Graduate Program during the collection of materials. The funding for the Article Publication Charge (APC) has been provided by the University of Lodz statutory funds. The funders had no role in study design, data collection and analysis, decision to publish, or preparation of the manuscript.

==============================
Background

Sexual dimorphism, driven by sexual selection, leads to varied morphological distinctions in male and female insects, providing insights into selection pressures across species. However, research on the morphometric variability within specific taxa of tiger beetles (Coleoptera: Cicindelidae), particularly arboreal and semi-arboreal species, remains very limited.

Methods

We investigate sexual dimorphism in six semi-arboreal Therates tiger beetle taxa from the Philippines, focusing on morphological traits. We employed morphometric measurements and multivariate analyses to reveal patterns of sexual dimorphism between sexes within the taxa.

Results

Our results indicate significant sexual dimorphism in elytra width, with females consistently displaying broader elytra, potentially enhancing fecundity. Notable sexual size dimorphism was observed in Therates fulvipennis bidentatus and T. coracinus coracinus, suggesting heightened sexual selection pressures on male body size. Ecological factors, mating behavior, and female mate choice might contribute to the observed morphological variation. These findings emphasize the need for further studies to comprehend mating dynamics, mate choice, and ecological influences on morphological variations in semi-arboreal and arboreal tiger beetles.

Introduction

Sexual dimorphism, widely observed in animals, involves morphological differences shaped by both sexual selection (Darwin, 1871; Andersson, 1994) and natural selection, as evidenced in adaptations such as feeding niche, predator avoidance, parental care roles, and environmental factors (e.g. Shine, 1989; Blanckenhorn, 2000, 2005; Cooper, Brown & Getty, 2015; Henshaw, Fromhage & Jones, 2019; Bauld et al., 2022). Insects often exhibit sexual size dimorphism (SSD), with females typically larger in body sizes due to fecundity selection (e.g. Blanckenhorn, 2005; Blanckenhorn, Meier & Teder, 2007; Stillwell et al., 2010; Rudoy & Ribera, 2017). Male-biased SSD is associated with mating success through male-male competition and female mate choice (Fairbairn & Preziosi, 1994; Blanckenhorn, 2005; Simmons & García-González, 2008; Benítez et al., 2013; Baur et al., 2020). These differences in sexual size and morphology have drawn significant interest in understanding the evolutionary implications of sexual dimorphism that influence the reproductive strategies and morphological characteristics of insects.

Tiger beetles (Coleoptera: Cicindelidae), a family of predatory insects (López-López & Vogler, 2017; Duran & Gough, 2020), are widely utilized as a model organism in biodiversity, ecology, and evolution, due to their adaptability to diverse environments, and their role as bioindicators of regional and global biodiversity patterns (e.g. Schultz, 1989; Knisley & Hill, 1992; Pearson & Cassola, 1992; Kitching, 1996; Carroll & Pearson, 1998a, 1998b; Andriamampianina et al., 2000; Cassola & Pearson, 2000; Pearson & Vogler, 2001; Pearson & Cassola, 2007; Jaskuła, 2011; Dangalle, Pallewatta & Vogler, 2014; Jaskuła, 2015; Jaskuła, Płóciennik & Schwerk, 2019; Pearson & Wiesner, 2023). However, most of these studies have been focused on epigeic ground-dwelling species. In the tropical regions, where tiger beetle species richness is the highest (Pearson & Wiesner, 2023), limited research has been conducted on their morphometric variability within specific species. Generally, it is observed that females are larger than males (Pearson & Vogler, 2001; Jaskuła, 2005; Espinoza-Donoso et al., 2020; Jaskuła, Schwerk & Płóciennik, 2021), with male-biased sexual size dimorphism distinctively observed among tiger beetles belonging to the Manticorini tribe (Mareš, 2002). Another sexual difference observed in almost all Cicindelidae (except genus Manticora) is the presence of tarsal adhesive setae on the ventral surfaces of the first three segments of the prothoracic tarsus in males, enabling them to better grip and secure females during copulation (Stork, 1980; Kritsky & Simon, 1995). Previous studies provided evidence of sexual dimorphism in tiger beetles, focusing on various morphological traits, such as the size and tooth arrangement of mandibles (Kritsky & Simon, 1995), mandible size and length (Matalin, 1999a, 1999b, 2002a, 2002b; Satoh et al., 2003; Satoh & Hori, 2004; Jaskuła, 2005; Ball, Acorn & Shpeley, 2011), elytral length (Franzen & Heinz, 2005), labrum (Cassola & Bouyer, 2007), and other body parameters (Jaskuła, 2005; Jaskuła, Schwerk & Płóciennik, 2021). Additionally, in certain genera, distinct sexual dimorphism has been observed in the shape of mandibles (Mareš, 2002; Jones & Conner, 2018), head and pronotum (Doğan Sarikaya, Koçak & Sarikaya, 2020), or abdomen and mandible (Espinoza-Donoso et al., 2020). Moreover, Jaskuła et al. (2016) combined phylogeographical analysis and morphological assessments to investigate sexual dimorphism and geographical variations in Calomera littoralis.

The genus Therates Latreille, 1817, comprised of almost 130 species distributed throughout Southeast Asia (Wiesner, 2020; Anichtchenko & Wiesner, 2022; Lin & Wiesner, 2022; Medina, Donato & Cabras, 2022; Matalin & Wiesner, 2023) are relatively small sized semi-arboreal tiger beetles with an unusual ecological niche for the adults that includes terrestrial and arboreal habitats. During the daytime, adults occur regularly on leaf surfaces of undergrowth forest vegetation, and occasionally on forest substrate, such as wooded paths and rocks along forest streams (Acal et al., 2021; Lin & Wiesner, 2022). Despite their prominence in southeast Asian forest, research on the morphometric variability of Therates has been limited.

The current understanding of sexual dimorphism in Cicindelidae from different regions of the world strongly indicates that females tend to be larger, primarily attributed to enhanced reproductive investment. For instance, a larger and wider abdomen in females potentially results in a higher egg production, signifying not only their reproductive role but also the need for additional energy and resources for finding suitable locations to lay eggs. In contrast, males typically invest only in sperm, leading to significantly lower energetic costs during courtship (Thornhill & Alcock, 1983). These aspects of mating behavior are commonly observed in Cicindelidae species. Moreover, tiger beetle males with larger mandibles may enhance copulation by facilitating better maintenance and grasping of the female’s thorax, as shown in previous studies (Pearson & Vogler, 2001).

Given the existing knowledge of sexual dimorphism in Cicindelidae from various regions worldwide, this study, the first part of a bigger project focused on the ecology of Philippine Therates, aims to investigate sexual dimorphism within particular taxa of this semi-arboreal tiger beetle genus. We hypothesize that akin to observations in other tiger beetles, females typically exhibit larger size compared to males, with the exception of single morphological parameters displaying higher values in males.

Materials and Methods

Sampling area

Mindanao is the second largest and southernmost major island in the Philippines with mean annual temperature of 80°F (27 °C), 77% relative humidity and an annual average rainfall of between 1,800–3,000 mm (Giles et al., 2019). Adult tiger beetles were collected from 49 sites on Mindanao and associated islands from November 2019 to January 2023 (Fig. 1, Table 1). The geographical position and altitude of each site were recorded using a Garmin eTrex 10 GPS. Prior informed consent and a Gratuitous Permit from Department of Environment and Natural Resources (DENR) were obtained for the collection. Wildlife Gratuitous Permits (WGP) numbered 319, R10 2019-81, R10 2019-48, R10 2019-89, R10 2021-38, along with Wildlife Export Permits (WEP) numbered R10-2021-04 and 2023-01, were secured. All samples were collected along riparian vegetation using an entomological hand net, immediately preserved in a 96% alcohol solution, and subsequently identified through taxonomic keys (Wiesner, 1988; Acal et al., 2021).

Figure 1 Geographical distribution and typical habitats of Philippine Therates tiger beetles.

(A) Philippine archipelago, (B) Geographical distribution of sampling sites in southern Philippines (blue dots on the map indicate precise sampling site locations; refer to Table 1 for site numbers and details, while colored contours represent varying elevations), (C) Selected habitats of Therates, (D) One of the studied arboreal tiger beetle species. Map & Photo credit: Dale Ann Acal.

Table 1 Sampling locations of Therates taxa in southern Philippines.

Sites	Region	Province	Species	GPS coordinates	Elevation (m a.s.l.)	
Latitude	Longitude	
1	Northern Mindanao	Lanao del Norte	i (64 ♂♂, 47 ♀♀)	8.22011	124.26780	72	
2	Northern Mindanao	Lanao del Norte	i (33 ♂♂, 13 ♀♀)	8.24896	124.42524	352	
3	Northern Mindanao	Lanao del Norte	i (10 ♂♂)	8.22625	124.29222	118	
4	Northern Mindanao	Lanao del Norte	i (45 ♂♂, 20 ♀♀)	8.27137	124.33088	119	
5	Northern Mindanao	Lanao del Norte	i (1 ♂, 1 ♀)	8.24251	124.46037	784	
6	Northern Mindanao	Lanao del Norte	i (6 ♂♂, 5 ♀♀)	8.27146	124.31414	116	
			iv (10 ♂♂, 5 ♀♀)				
7	Northern Mindanao	Misamis Oriental	i (1 ♂, 1 ♀)	8.39697	124.39840	433	
			iv (3 ♂♂, 2 ♀♀)				
8	Northern Mindanao	Lanao del Norte	i (20 ♂♂, 7♀♀)	8.20788	124.28481	163	
			iv (3 ♂♂, 7 ♀♀)				
9	Northern Mindanao	Lanao del Norte	i (16 ♂♂, 6 ♀♀)	8.24944	124.37917	322	
			iv (21 ♂♂, 13 ♀♀)				
10	Northern Mindanao	Lanao del Norte	i (26 ♂♂, 9 ♀♀)	8.27938	124.33738	212	
			iv (26 ♂♂, 9 ♀♀)				
11	Northern Mindanao	Bukidnon	i (5 ♂♂)	7.91148	124.81047	1,546	
			v (4 ♂♂, 1 ♀)				
12	Zamboanga Peninsula	Zamboanga del Sur	i (3 ♂♂)	7.05121	121.98278	354	
			v (5 ♂♂, 2 ♀♀)				
13	Zamboanga Peninsula	Zamboanga del Sur	i (5 ♂♂, 1 ♀)	7.14045	121.92201	93	
			v (2 ♂♂, 2 ♀♀)				
14	Zamboanga Peninsula	Zamboanga del Sur	i (17 ♂♂, 6 ♀♀)	7.08457	121.91796	185	
			v (25 ♂♂, 4 ♀♀)				
15	Northern Mindanao	Misamis Oriental	ii (2 ♂♂)	8.94884	124.86517	501	
16	Northern Mindanao	Misamis Oriental	ii (4 ♂♂, 2 ♀♀)	8.68667	125.00500	1,170	
17	Northern Mindanao	Misamis Oriental	ii (15 ♂♂, 4 ♀♀)	8.69333	125.00944	1,190	
18	Davao	Davao del Sur	ii (37 ♂♂, 32 ♀♀)	7.11229	125.31761	1,218	
19	Davao	Davao del Sur	ii (5 ♂♂, 3 ♀♀)	7.23326	125.39823	290	
			v (6 ♂♂, 1 ♀)				
20	CARAGA	Surigao del Sur	ii (17 ♂♂, 18 ♀♀)	8.14263	126.25162	21	
			iii (4 ♂♂, 2 ♀♀)				
			vi (3 ♂♂)				
21	Northern Mindanao	Bukidnon	ii (2 ♂♂)	8.25632	125.17292	1,142	
			iii (1 ♂, 2 ♀♀)				
			v (31 ♂♂, 19 ♀♀)				
22	Northern Mindanao	Bukidnon	ii (2 ♂♂)	8.22175	125.33145	721	
			iii (1 ♂)				
			v (2 ♀♀)				
23	CARAGA	Agusan del Sur	ii (17 ♂♂, 14 ♀♀)	8.55795	125.98071	104	
			vi (18 ♂♂, 13 ♀♀)				
24	CARAGA	Agusan del Sur	ii (15 ♂♂, 28 ♀♀)	8.66492	126.01571	115	
			vi (9 ♂♂, 3 ♀♀)				
25	CARAGA	Surigao del Sur	ii (2 ♂♂, 7 ♀♀)	8.56033	126.03343	108	
			vi (1 ♂)				
26	Northern Mindanao	Bukidnon	ii (1 ♂,)	7.90445	124.81776	1,342	
			v (3 ♂♂)				
27	CARAGA	Surigao del Sur	iii (15 ♂♂, 9 ♀♀)	8.13729	126.30099	48	
			vi (36 ♂♂, 20 ♀♀)				
28	CARAGA	Agusan del Sur	iii (6 ♂♂, 5 ♀♀)	8.48556	125.99369	153	
			vi (5 ♂♂, 2 ♀♀)				
29	CARAGA	Surigao del Sur	iii (2 ♂♂)	8.17222	126.22861	70	
			vi (9 ♂♂, 3 ♀♀)				
30	CARAGA	Surigao del Sur	iii (7 ♂♂, 3 ♀♀)	8.14339	126.26574	60	
			vi (6 ♂♂, 3 ♀♀)				
31	CARAGA	Surigao del Sur	iii (5 ♂♂)	9.28161	125.88619	80	
			vi (1 ♂, 1 ♀)				
32	Northern Mindanao	Misamis Oriental	iv (15 ♂♂, 10 ♀♀)	8.38761	124.38864	357	
33	Northern Mindanao	Bukidnon	iv (2 ♂♂)	7.91140	124.81145	1,342	
34	Zamboanga Peninsula	Zamboanga del Sur	iv (11 ♂♂, 3 ♀♀)	7.88446	123.42919	175	
35	Zamboanga Peninsula	Zamboanga del Sur	iv (11 ♂♂, 8 ♀♀)	7.75187	123.29394	130	
36	Northern Mindanao	Misamis Oriental	iv (4 ♂♂, 8 ♀♀)	8.94884	124.86517	501	
37	Zamboanga Peninsula	Zamboanga del Sur	v (6 ♂♂, 2 ♀♀)	8.26250	123.53703	728	
38	Northern Mindanao	Lanao del Norte	v (1 ♂, 1 ♀)	8.24546	124.45381	914	
39	Northern Mindanao	Lanao del Norte	v (6 ♂♂, 2 ♀♀)	8.23577	124.44924	885	
40	Northern Mindanao	Lanao del Norte	v (3 ♂♂, 1 ♀)	8.24546	124.45381	914	
41	Northern Mindanao	Bukidnon	v (3 ♂♂, 1 ♀)	8.22175	125.33145	721	
42	Northern Mindanao	Camiguin island	v (2 ♂♂)	9.19274	124.68340	693	
43	CARAGA	Dinagat Island	vi (3 ♂♂, 2 ♀♀)	10.3166	125.58540	22	
44	CARAGA	Agusan del Sur	vi (10 ♂♂, 5 ♀♀)	8.69922	125.97482	170	
45	CARAGA	Surigao del Sur	vi (8 ♂♂, 3 ♀♀)	8.16111	126.29319	53	
46	CARAGA	Surigao del Sur	vi (2 ♂♂, 1 ♀)	8.16111	126.29319	625	
47	CARAGA	Surigao del Sur	vi (5 ♂♂)	8.16111	126.29319	52	
48	CARAGA	Surigao del Sur	vi (17 ♂♂, 9 ♀♀)	9.20268	126.00482	180	
49	CARAGA	Surigao del Sur	vi (2 ♂♂)	9.26596	125.89413	126	
Note:

(i) T. fulvipennis bidentatus, (ii) T. coracinus coracinus, (iii) T. fulvipennis everetti, (iv) T. fasciatus fasciatus, (v) T. fasciatus pseudolatreillei, and (vi) T. fasciatus quadrimaculatus.

Morphometric measurements

To determine morphological variations of Therates taxa, we measured relevant morphological characteristics of all individuals. In total, 1,176 specimens were studied (762♂♂ 418♀♀) from the following species: T. fulvipennis bidentatus Chaudoir, 1861 (253♂♂ 116♀♀), T. coracinus coracinus Erichson, 1834 (118♂♂ 108♀♀), T. fulvipennis everetti Bates, 1878 (40♂♂ 21♀♀), T. fasciatus fasciatus Fabricius, 1801 (113♂♂ 69♀♀), T. fasciatus pseudolatreillei Horn, 1928 (100♂♂ 37♀♀), and T. fasciatus quadrimaculatus Horn, 1895 (135♂♂ 66♀♀). Quantitative measurements, adapted from Jaskuła (2005) were as follows (Fig. 2): right mandible length (RML), length of head (LH), width of head (WH), width of pronotum (WP), length of pronotum (LP), length of elytra (EL), width of elytra (WE), and total body length (TBL). Additionally, the analysis included labrum length (LL), and specimen weight (WT) in grams. The sex was determined by examining the abdominal apex and forelegs, the latter with presence of the tarsal adhesive setae located on the first three tarsi of the prothoracic leg of males (Pearson, 1988; Stork, 1980). A Nikon SMZ800 stereoscope with micrometric ocular measurement (Delta optical DLTA12000CM0SSEU3) was used to measure the morphological features in millimeters and KERN PCB precision balance was used to measure beetle’s weight.

Figure 2 Graphical representation of Therates with morphometric parameters.

(1) Right mandible length (RML), (2) labrum length (LL), (3) length of head (LH), (4) width of head (WH), (5) length of pronotum (LP), (6) width of pronotum (WP), (7) length of elytra (EL), (8) width of elytra (WE), and (9) total body length (TBL). Credit: Dale Ann Acal.

Data analysis

All body parameters were standardized against total body length by dividing the measured values by the total body length for each individual. These standardized values were compared for males and females of all six taxa. In many instances, the Kolmogorov-Smirnov test rejected normal distribution of the data. Therefore, we assessed the statistically significant differences between the sexes within each population using non-parametric Mann–Whitney U tests. Box plots were constructed with confidence intervals for total body length, elytral width and beetle’s weight using R packages: ggplot2 (Wickham, 2016) and dplyr (Wickham et al., 2020). Subsequently, principal component analysis (PCA) was used to determine morphological parameters responsible for the variation between sexes of each Therates taxa. Additionally, a one-way permutation multivariate analysis of variance (PERMANOVA) was conducted to examine the differences on the observed variations in morphological traits within each taxa. All statistical analyses were performed using RStudio software (v. 5.2.4; RStudio Team, 2024).

Results

The results of the one-way PERMANOVA revealed a highly significant differences in morphological parameters between sex of T. fulvipennis bidentatus (F = 39.46, df = 1, p < 0.001), T. coracinus coracinus (F = 24.54, df = 1, p < 0.001), T. fulvipennis everetti (F = 3.42, df = 1, p < 0.009), T. fasciatus fasciatus (F = 19.91, df = 1, p < 0.001), T. fasciatus pseudolatreillei (F = 17.61, df = 1, p < 0.001), and T. fasciatus quadrimaculatus (F = 19.22, df = 1, p < 0.001) (Table 2). The PCA results revealed that the first two principal components collectively explained 68.9% of the total morphological variation. In T. fulvipennis bidentatus, PC1 was mainly influenced by right mandible length (RML = 0.47), labrum length (LL = 0.51), and elytral length (EL = −0.5) while PC2 was influenced by head length (LH = −0.59), pronotum length (LP = 0.59) and width (WP = 0.41). Subsequently, for T. coracinus coracinus, right mandible length (RML = −0.41), elytral width (WE = 0.44) and length (EL = 0.5) contributed significantly to the observed variations in PC1 and pronotum length (LP = 0.59) and width (WP = 0.64) in PC2. In the case of T. fulvipennis everetti, width of head (WH = −0.53), pronotum length (LP = 0.52) and width (WP = 0.48) contributed to the variation observed in PC1, while elytral length (EL = 0.53), right mandible length (RML = −0.54), and labrum length (LL = −0.59) contributed mostly in PC2. For T. fasciatus fasciatus, labrum length (LL = −0.42), elytral width (WE = 0.46) and length (EL = 0.52) were the main factors to the variations in PC1, whereas pronotum length (LP = 0.58) and width (WP = 0.53) contributed significantly in PC2. In T. fasciatus pseudolatreillei, elytral width (WE = 0.44) and length (EL = 0.49) were the main factors to the variations observed in PC1 while pronotum length (LP = 0.58) and width (WP = 0.64) in PC2. Lastly, in T. fasciatus quadrimaculatus, pronotum width (WP = 0.4), head length (LH = 0.44) and width (WH = −0.44) were the primary factors in the observed variation in PC1, while PC2 was mainly delimited by elytral length (EL = 0.7) (Fig. 3).

Table 2 One-way PERMANOVA results obtained by examining the morphological differences between sexes in each taxon (9,999 permutation, Euclidean distance).

Species	df	Sum of Sqs.	R2	F	P	
T. fulvipennis bidentatus	1	0.009109	0.097086	39.46165	0.001	
Residual	367	0.084717	0.902914			
Total	368	0.093826	1			
T. coracinus coracinus	1	0.005809	0.098719	24.53524	0.001	
Residual	224	0.05303	0.901281			
Total	225	0.05884	1			
T. fulvipennis everetti	1	0.000824	0.054723	3.415538	0.009	
Residual	59	0.014234	0.945277			
Total	60	0.015058	1			
T. fasciatus fasciatus	1	0.005994	0.099579	19.9065	0.001	
Residual	180	0.054199	0.900421			
Total	181	0.060193	1			
T. fasciatus pseudolatreillei	1	0.004864	0.115395	17.6104	0.001	
Residual	135	0.037286	0.884605			
Total	136	0.04215	1			
T. fasciatus quadrimaculatus	1	0.00616	0.08809	19.2237	0.001	
Residual	199	0.06375	0.91190			
Total	200	0.06991	1			

Figure 3 Principal component analysis biplot for morphometric parameters between sexes for each taxon.

(i) T. fulvipennis bidentatus, (ii) T. coracinus coracinus, (iii) T. fulvipennis everetti, (iv) T. fasciatus fasciatus, (v) T. fasciatus pseudolatreillei, and (vi) T. fasciatus quadrimaculatus. Arrows represent the measured variables (See Fig. 2 for trait abbreviations): blue—males, red—females.

Significant size differences were observed between males and females in Therates fulvipennis bidentatus and T. coracinus coracinus, with males being larger. Although males in all three T. fasciatus subspecies exhibited higher median values than females, these differences were not statistically significant. In T. fulvipennis everetti, the median value for females was higher than that of males, but the sample size was inadequate for conclusive results. Analysis of standardized values for all six taxa revealed significant differences in elytral width (WE). Median values indicated that females are wider than males across all studied Therates taxa. Right mandible length (RML) showed higher median values in males for all taxa, with significant sexual differences in three taxa (T. fulvipennis bidentatus, T. coracinus coracinus and T. fasciatus quadrimaculatus). Additionally, weight comparison between sexes revealed no statistically significant differences, except for T. fulvipennis bidentatus and T. fasciatus quadrimaculatus, where males had significantly higher median values (Fig. 4).

Figure 4 Total body length, elytral width, weight, and right mandible length between males and females of the studied Therates.

(i) T. fulvipennis bidentatus, (ii) T. coracinus coracinus, (iii) T. fulvipennis everetti, (iv) T. fasciatus fasciatus, (v) T. fasciatus pseudolatreillei, and (vi) T. fasciatus quadrimaculatus. The boxes represent the interquartile range, with the median shown as the horizontal line within the box. See Fig. 2 for trait abbreviations. Mann–Whitney U tests between sexes in each taxon (*p < 0.001).

Discussion

The influence of sexual selection on morphological traits is widely recognized, but the direction and degree of sexual size dimorphism vary greatly across species and populations (e.g. Andersson, 1994; Blanckenhorn, Meier & Teder, 2007; Stillwell et al., 2010). This study provides an essential contribution to our understanding of sexual dimorphism in semi-arboreal tiger beetles as well as for tropical Cicindelidae. In contrast to the prevailing pattern observed in most insects and ground-dwelling tiger beetles, where females generally exhibit larger body size than males (Pearson & Vogler, 2001; Jaskuła, 2005; Jaskuła, Schwerk & Płóciennik, 2021), our results reveal significant sexual dimorphism in Therates taxa, but with males larger than females in two species and about the same size in other taxa. A comparison of total body length between sexes showed clear differences in T. fulvipennis bidentatus and T. coracinus coracinus, while the patterns are less evident among all studied subspecies of T. fasciatus. Although females were approximately the same size as males in most studied taxa, T. fulvipennis bidentatus and T. coracinus coracinus exhibited clear sexual size dimorphism with males larger. Such male-biased SSD is relatively uncommon in beetles, with only 9% of reported species exhibiting this pattern, compared to 72% with females larger (Stillwell et al., 2010). Among tiger beetles, male-biased sexual size dimorphism was only observed in species belonging to the Manticorini tribe (Mareš, 2002). The larger size of males in Therates may be a derived trait resulting from an increase in the intensity of sexual selection on male body size. It is widely documented that larger males have a competitive advantage in intrasexual competition, where they engaged in competitive behavior with other males to gain access to females (Thornhill & Alcock, 1983; Andersson, 1994; Emlen, 1997; Fincke, Waage & Koenig, 1997; Benítez et al., 2013). In such situations, larger size provided males with increased strength and fighting ability to outcompete smaller rivals and secure mating opportunities (Andersson, 1994; Emlen, 1997; Blanckenhorn, 2005; Stillwell et al., 2010). Mate-guarding behavior, observed in various tiger beetle species, involves males staying attached to females after mating (Pearson & Vogler, 2001). This behavior prevents other males from mating with the female, increasing the male’s chances of fertilizing her eggs. As mating occurs in the vicinity where females deposit their eggs, and females typically remain in this area for an extended period, there is an increasing potential for male-male competition (Alcock, 1994; Tigreros & Kattan, 2008). In previous study of the mating behavior of tropical Cicindela s.l. species, Shivashankar & Pearson (1994) reported prolonged copulations, where males tend to remain mounted on females for extended durations in the presence of competitors. Additionally, they observed a higher proportion of males compared to females in the mating area, suggesting a potential male-male competition for access to females. Our present study also found a higher proportion of males, consistent with the earlier findings, which provides additional support for the possible existence of male-male competition during mating encounters in this taxonomic group. Moreover, female mate choice may influence male tiger beetles body size, as larger males are often preferred by females in various insect groups (e.g. Emlen, 1997), indicating good health, genetic quality, or superior fitness. This provides an indirect benefit for females’ offspring, such as enhanced survival or improved genetic traits (Darwin, 1871; Neff & Pitcher, 2005; Fedina & Lewis, 2007). Tiger beetles exhibit pre-mating struggles, and females may assess males’ abilities during these encounters, suggesting a potential mate choice mechanism (Pearson, 1988; Tigreros & Kattan, 2008).

While our study identified clear sexual size dimorphism in two Therates species, T. fulvipennis bidentatus and T. coracinus coracinus, where males were larger than females, we also found that in three studied subspecies of T. fasciatus, males were approximately the same size as females. This may suggest potential variations in factors like mating behaviour, ecological pressures, or selective pressures on male body size among studied taxa. While our understanding of the life history, ecology, and particularly the behavior of Therates species is limited, it is plausible that some males employ different strategies during courtship. Alternative mating strategies, observed in other tiger beetle genera like Pseudoxycheila (Tigreros & Kattan, 2008), might similarly contribute to phenotypic variation in semi-arboreal Therates. Mating with females ready to oviposit could be advantageous for smaller males if last-male sperm precedence occurs, enabling them to exploit these opportunities and contribute to the observed phenotypic variation in male body size (Thornhill & Alcock, 1983; Alcock, 1994; Shivashankar & Pearson, 1994).

Although the evolution in male body size variance in tiger beetles is poorly known, in some other beetle groups the evolution of sexual size dimorphism is mainly influenced by changes in males. The higher evolutionary variance in male body size may be due to a lack of constraints associated with egg development and reproduction in females. This absence of constraints could drive the evolution of male body size, leading to minimized males in some species (e.g. Rudoy & Ribera, 2017). Moreover, the occurrence of similar-sized males and females may be driven by ecological factors such as specialized feeding behaviors or foraging strategies, potentially as a mechanism to avoid intraspecific resource competition. However, evidence supporting sexual size dimorphism primarily originating from ecological divergence is limited across different groups (e.g. Slatkin, 1984; Shine, 1989; Fairbairn, 1997; Mysterud, 2000; Vincent & Herrel, 2007). Furthermore, underlying ecological factors may also influence sexual size dimorphism in tiger beetles. Prey availability during the larval stages is a critical determinant of adult body size in insects, including tiger beetles (Pearson & Knisley, 1985). Larval competition for prey could result in smaller adult body sizes, particularly when food resources are limited (Stillwell et al., 2010).

Consistent with previous research on tiger beetles (Pearson & Vogler, 2001; Franzen & Heinz, 2005; Jaskuła, 2005; Jaskuła, Płóciennik & Schwerk, 2019; Espinoza-Donoso et al., 2020; Jaskuła, Schwerk & Płóciennik, 2021) and other insects (e.g. Thornhill & Alcock, 1983; Fairbairn, 1997; Stillwell et al., 2010), our study found a prominent sexual dimorphism in elytral width, with females consistently exhibiting broader elytra than males across all six studied taxa. This disparity in elytra is likely related to the greater parental investment by females (Thornhill & Alcock, 1983; Pearson & Vogler, 2001), as broader elytra usually mean a wider abdomen shape, providing additional space for developing and carrying a larger number of eggs, potentially enhancing their fecundity. Recent findings of a wider abdomen shape in females of Cicindelidia trifasciata by Espinoza-Donoso et al. (2020) strongly support this hypothesis. This suggested a potential protective role for developing eggs, possibly for better insulation and protection against external threats, such as predators or adverse environmental conditions (Goczał & Beutel, 2023). In addition to the notable sexual dimorphism in elytra width, variations in size and shape of mandibles and labrum between males and females are also important morphological traits due to their potential role in courtship and copulation dynamics (Pearson & Vogler, 2001). Our findings showed that males have longer mandibles than females in three taxa: Therates fulvipennis bidentatus, T. coracinus coracinus, and T. fasciatus quadrimaculatus. These findings confirm those of earlier studies, on European (Matalin, 1999a, 1999b, 2002a, 2002b; Jaskuła, 2005), South American (Espinoza-Donoso et al., 2020), and African taxa (Mareš, 2002; Ball, Acorn & Shpeley, 2011; Jaskuła, Schwerk & Płóciennik, 2021). Larger mandibles in males of these species are likely associated with increased mating success, providing better grip during copulation (Pearson & Vogler, 2001; Jaskuła, 2005). The investigation of mandibular length’s role in structuring tiger beetle communities, particularly through resource partitioning, further emphasizes its significance in this taxon (Pearson & Mury, 1979; Pearson & Vogler, 2001; Satoh et al., 2003; Satoh & Hori, 2004). Additionally, mandible size may also influence prey selection during hunting behavior, as observed in other Cicindelidae species (Gilbert, 1997; Rewicz & Jaskuła, 2018). However, contrary to what has been reported by Kritsky & Simon (1995) and Cassola & Bouyer (2007), labra from four Therates taxa were observed to be longer in males than females, except T. fulvipennis everetti, and T. fasciatus pseudolatreillei. Furthermore, in all the studied taxa, except for T. fulvipennis everetti, females exhibited a shorter and narrower pronotum compared to males. Similar results were observed in Cephalota circumdata cappadocica, where females had shorter pronotum, which, although wider than in males, showed a slightly narrower posterior part (Doğan Sarikaya, Koçak & Sarikaya, 2020). This sexual dimorphism in pronotal morphology can be associated with reproductive functions. During mating in tiger beetles, males engaged in a distinct behaviour, gripping the female’s thorax with their mandibles. This behaviour was facilitated by specific grooves (coupling sulci), located on the rear part of the female’s thorax, providing a secure hold for the male’s mandibles. The species-specific morphology of these grooves affected mandible compatibility, influencing the likelihood of successful mating, as documented in previous studies (Freitag, 1974; Shivashankar & Pearson, 1994).

Conclusions

This study highlights the general patterns of sexual dimorphism observed in semi-arboreal Therates. Our results reveal significant sexual dimorphism in elytra width in all studied taxa. Females consistently exhibit broader elytra than males, potentially enhancing fecundity through increased egg-carrying capacity. Males, particularly in Therates fulvipennis bidentatus and T. coracinus coracinus, exhibit male-biased sexual size dimorphism, indicating potential increased sexual selection pressures. Further research is needed to examine the functional roles of these morphological traits in semi-arboreal and arboreal tiger beetles particularly in relation to mating behavior, mate choice patterns, and the ecological factors influencing morphological variations. In addition, future studies should aim to expand the sample size and incorporate genetic analyses from different geographic regions to gain a more comprehensive understanding of the adaptive significance of sexual dimorphism in semi-arboreal tiger beetles.

Supplemental Information

Supplemental Information 1 Raw measurements, including all morphological traits for the studied Therates taxa.

The first author thanks the following persons for their help in the field: Angel Luz Lisondra, James Lucas, Sam Anthony Acal, Gypsie Claudine Café, Obed Bagona, Rogie Madula, Chrystel Mae Correos, Lito Mancao, Jose Chan, Wilson Cacho, Kem Leovie Ruste and Darell Mecca. We thank Olga Nuñeza for her kind help in organizing the 2019 Philippine expedition and Andrea Desiderato for invaluable statistical guidance. We acknowledge the contribution made by Noville Jay Ebina, Marco Luis Lumontod, Princess Johara Malawani, and Johanna Aseniero for enabling the study of select Therates materials collected under their efforts. Finally, we thank Barry Knisley, Alexandre Palaoro, and two anonymous reviewers for their helpful comments that significantly improved the manuscript.

Additional Information and Declarations

Competing Interests

Author Contributions

Field Study Permissions

Data Availability

The authors declare that they have no competing interests.

Dale Ann Acal conceived and designed the experiments, performed the experiments, analyzed the data, prepared figures and/or tables, authored or reviewed drafts of the article, and approved the final draft.

Anna Sulikowska-Drozd analyzed the data, authored or reviewed drafts of the article, and approved the final draft.

Radomir Jaskuła conceived and designed the experiments, analyzed the data, authored or reviewed drafts of the article, and approved the final draft.

The following information was supplied relating to field study approvals (i.e., approving body and any reference numbers):

Permissions to conduct field collections were obtained from the Department of Environment and Natural Resources - Biodiversity Management Bureau (DENR-BMB) through a Wildlife Gratuitous Permits (WGP) no. 319, DENR- Region 10, WGP no. R10 2019-81, WGP no. R10 2019-48 (Mt. Kalatungan Range Natural Park), WGP no. R10 2019-89 (Camiguin Island), WGP no. R10 2021-38 (Renewal) (Northern Mindanao) and Wildlife Export Permit (WEP) no. R10-2021-04, WEP no. 2023-01.

The following information was supplied regarding data availability:

The raw measurements, including all morphological traits for the studied taxa, are available in the Supplemental File.

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
