# Peer review of "Filling the gaps in ecology of tropical tiger beetles (Coleoptera: Cicindelidae): first quantitative data of sexual dimorphism in semi-arboreal Therates from the Philippine biodiversity hotspot"

_PeerJ, doi:10.7717/peerj.16956_

## Round 0.1 · original submission · Major Revisions

Dear Dr. Acal,

After this first review round, three reviewers decided that minor reviews were needed in your manuscript. Still, one last reviewer recommended rejecting it. Although the "minor-review" reviewers provided valuable insights, the "reject" reviewer raised important issues that prevent the acceptance of your manuscript in its current form. Considering all suggestions made by the reviewers, I believe a significant major review of your manuscript is necessary to make your manuscript scientifically meaningful. Therefore, please consider changing your text to cover all issues raised by the reviewers, especially those suggestions made by reviewer #4, who rejected your manuscript. Of course, it is possible to reject those changes. Still, you must provide compelling arguments to convince the above reviewer. On the other hand, if you can provide significant changes in the direction suggested by that reviewer and if they agree the next version of your manuscript is scientifically correct, I am sure you will not have trouble having the manuscript accepted for publication in PeerJ. Please also note that it is not mandatory that you include the citations requested by Reviewer 2.

So, to summarize, your manuscript may be accepted for publication in Peerj after you prepare a new version of your text.

Sincerely,
Daniel Silva

·

Basic reporting

This is a generally well designed and well written manuscript. I do not include any grammar edits or other significant suggested changes as writing is good. The introduction is very complete with relevant literature included. The authors state that in tiger beetles, it is generally know that females are larger. True bu they should confirm if there are any reports of males being larger in any species. The methods are clearly states and the statistical analysis seems appropriate. As stated, this contribution on the genus Therates is worthwhile and adds needed info on this genus.

Experimental design

Design is appropriate with sufficiently large numbers of specimens measured and statistical analysis suitable for the stated objectives of the study

Validity of the findings

While the study is not especially novel, it does expand the study of sexual dimorphism to additiona taxa, so that is valuable. Also significant is the confirmation of larger males in this group of insects, and this is novel.
The discussion provides an excellent review of factors that support the advantages of larger male size. But, I also think the authors should address why only 2 of species had larger males….any biological or other explanations. Or, is it a statistical thing that the males in other species were larger but not significantly larger. It would seem that the advantages of larger males size in these closely related? Species would apply to all of the species studied. A comment on differences in mandible size. I would think that if larger mandible size was important in structuring this community as suggested based on other studies, both males and females should have larger mandibles (same size). I would think the larger mandibles in the males of these species in more likely related to mating success (better holding females in copulation). Larger size and larger mandibles could result in more successful mating.

Reviewer 2 ·

Basic reporting

lines 66-68
Please, indicate the particular species for which the combined phylogeographical analysis and morphological assessments to investigate sexual dimorphism and geographical variations was conducted by Jaskula et al. (2016).

lines 75, 76
Despite their prominence in southeast Asian forest, research on the morphometric variability of Therates is lacking.
It's not entirely true. Please, see Matalin & Wiesner, 2023.

Some reference sources are ignored.
lines 61, 62
original version - mandible size and length (Satoh et al., 2003; Satoh & Hori, 2004; Ball, Acorn & Shpeley, 2011)
corrected version - mandible size and length (Matalin, 1999a,b, 2002a, b; Satoh et al., 2003; Satoh & Hori, 2004; Ball, Acorn & Shpeley, 2011)

lines 228, 229
original version - These findings confirm those of earlier studies, on European (Jaskula, 2005), ...
corrected version - These findings confirm those of earlier studies, on European (Matalin, 1999a,b; 2002a,b; Jaskula, 2005), ...

Four literature sources could be added in the References:
Matalin A.V., 1999a. The tiger-beetles of the 'hybrida' species-group. II. A taxonomic review of subspecies in Cicindela sahlbergii Fischer von Waldheim, 1824 (Coleoptera, Carabidae, Cicindelinae). P. 13-55 / In book (Eds. A. Zamotailov & R. Sciaki): “Advanced in Carabidology: Papers dedicated to the memory of Professor Oleg L. Kryzhanovskij”, MUISO Publ.
Matalin A.V., 1999b. Taxonomic status and subspecies structure of Cicindela altaica (Coleoptera, Carabidae) // Entomological Review, Vol. 79, No. 7, 1999, P. 809–820 [translated from Zoologicheskii Zhurnal, Vol. 78, No. 5, 1999].
Matalin A.V., 2002a. The taxonomic status of Cicindela (Cicindela) reitteri Horn, 1897, with some remarks about the subspecies and forms of Cicindela (Cicindela) maritima Dejean, 1822 (Coleoptera: Carabidae, Cicindelinae) - Teil 1 // Entomologische Zeitschrift, Stuttgart, Jahr. 112, Nr. 3, S. 89-94.
Matalin A.V., 2002b. The taxonomic status of Cicindela (Cicindela) reitteri Horn, 1897, with some remarks about the subspecies and forms of Cicindela (Cicindela) maritima Dejean, 1822 (Coleoptera: Carabidae, Cicindelinae) – Teil 2 // Entomologische Zeitschrift, Stuttgart, Jahr. 112, Nr. 4, S. 98-107.

Experimental design

No comments.

Validity of the findings

lines 168-170
This study provides the first quantitative report of sexual dimorphism in semi-arboreal tiger beetles as well as for tropical Cicindelidae.
It's not entirely true. Please, see Matalin & Wiesner, 2023 and take in into account.

Additional comments

No comments.

Reviewer 3 ·

Basic reporting

The basis for this study and its significance is straight forward and readily comprehended,

Experimental design

Lines 78-81: The hypothesis and predictions to be tested could be made clearer here, and in so doing would considerably clarify why you measured what you did.How does the stated hypothesis for these semi-arboreal species differ from what you might expect for ground-dwelling species and why? Would the different behaviour of the semi-arboreal species make sexual dimorphism more or less obvious and/or consistent than that of terrestrial species? Your results show that some semi-arboreal species follow terrestrial species in having females larger than males, but other species are just the opposite - something rarely recorded among tiger beetles.Could you have predicted this from factors unique for semi-arboreal species? What is different about the semi-arboreal species where males are large than females from those congeners where females are larger than males? Again including these factors in you introductory hypothesis and predictions would stregnthen the scientific method applied.

Validity of the findings

Strong statistics and justified results.

·

Basic reporting

I am sending a full review attached (it has a graph in it), but I feel that the paper misses a good portion of the literature by not considering that sexual dimorphism can emerge from natural selection. I add on that point in the attached document.

Experimental design

I feel that the analyses are missing important points of working with allometry (which is a major goal of this paper). PCAs and ratios are not sufficient to show how much species differ regarding proportionality and investment.

Validity of the findings

The data is robust and there is a lot of it! However, as above, I feel that the statistics are not properly defined.

Additional comments

I am sending a docx comment with a full review. I sincerely hope it helps.

---

## Round 0.2 · Minor Revisions

Dear Dr. Acal,

After this new review round, all reviewers were very positive regarding your manuscript. They all decided on an acceptance of your manuscript or minor reviews. One of the reviewers suggested that the sampling data in Table 1 should also be added to Figure 1. The number of males and females captured in each sampling site should also be shown. Another reviewer suggested that a minor rephrasing of the hypotheses of the manuscript should be performed before it is accepted for publication.

Sincerely,
Daniel Silva

·

Basic reporting

No issues, well written

Experimental design

Design and analysis is good

Validity of the findings

Good

Additional comments

I have completed a complete review of this previously reviewed manuscript and find all of the suggested changes have been adequately addressed, and now find the manuscript acceptable for publication

Reviewer 2 ·

Basic reporting

For better clarity, the numbers of the sampling sites from Table 1 could be placed in Figure 1B (as numbers in each red circle).
It is advisable to add to Table 1 the number of males and females of each species captured at each sampling site (in parentheses after the appropriate Roman numeral).

Experimental design

No comments

Validity of the findings

No comments

Reviewer 3 ·

Basic reporting

No comment

Experimental design

There is a confusion of what constitues an hypothesis. Some rewording of the goals of this study are indicated on the review comments attached.to the revised text.

Validity of the findings

No comment

Annotated reviews are not available for download in order to protect the identity of reviewers who chose to remain anonymous.

---

## Round 0.3 · accepted · Accept

Dear Dr. Acal,

I am pleased to inform you that your manuscript has been accepted for publication in PeerJ! Congratulations!

Sincerely,
Daniel Silva

Reviewer 2 ·

Basic reporting

No comments

Experimental design

No comments

Validity of the findings

No comments

Reviewer 3 ·

Basic reporting

Clear

Experimental design

Well-defined

Validity of the findings

Documents objectively quantify sexual dimorphism among tiger beetles..

Additional comments

The autors have worked out many suggested changes and details that improve the ms. and its message.